# Resveratrol as a Multifunctional Topical Hypopigmenting Agent

**DOI:** 10.3390/ijms20040956

**Published:** 2019-02-22

**Authors:** Jung-Im Na, Jung-Won Shin, Hye-Ryung Choi, Soon-Hyo Kwon, Kyung-Chan Park

**Affiliations:** Department of Dermatology, Seoul National University Bundang Hospital, Seongnam 463-707, Korea; vividna@gmail.com (J.-I.N.); spellbound00@hanmail.net (J.-W.S.); hrchoi73@snu.ac.kr (H.-R.C.); soonhyo17@hanmail.net (S.-H.K.)

**Keywords:** resveratrol, hypopigmenting agent, melanogenesis, tyrosinase inhibitor, antioxidant

## Abstract

Melanin is produced in melanocytes and stored in melanosomes, after which it is transferred to keratinocytes and, thus, determines skin color. Despite its beneficial sun-protective effects, abnormal accumulation of melanin results in esthetic problems. A range of topical hypopigmenting agents have been evaluated for their use in the treatment of pigmentary disorders with varying degrees of success. Hydroquinone (HQ), which competes with tyrosine, is the main ingredient in topical pharmacological agents. However, frequent occurrence of adverse reactions is an important factor that limits its use. Thus, efforts to discover effective topical hypopigmenting agents with less adverse effects continue. Here, we describe the potential of resveratrol to function as an effective hypopigmenting agent based on its mechanism of action. Resveratrol is not only a direct tyrosinase inhibitor but an indirect inhibitor as well. Additionally, it can affect keratinocytes, which regulate the function of melanocytes. Resveratrol regulates the inflammatory process of keratinocytes and protects them from oxidative damage. In this way, it prevents keratinocyte-induced melanocyte stimulation. Furthermore, it has a rescuing effect on the stemness of interfollicular epidermal cells that can repair signs of photoaging in the melasma, a typical pigmentary skin disorder. Overall, resveratrol is a promising potent hypopigmenting agent.

## 1. Introduction

Melanin plays an important role in protecting the skin from ultraviolet (UV) light. It also determines skin color and influences phenotypic appearances. However, increased levels of melanin may lead to esthetic problems. Hydroquinone (HQ) is the main ingredient in topical pharmacological agents used for the treatment of hyperpigmentary disorders. However, HQ is frequently associated with a high occurrence of adverse effects, including contact dermatitis, irritation, and burning/prickling sensations [1,2]. It can also infrequently induce serious problems, including leukoderma, hypochromia, and ochronosis [3,4]. Therefore, several topical hypopigmenting agents with less adverse effects have been developed [5]. Table 1 shows these hypopigmenting agents classified according to their mechanisms of action.

Resveratrol is a natural polyphenol present in various fruits and vegetables. It has been proven to play an important role as a therapeutic and chemopreventive agent used in the treatment of various illnesses, including cancers and cardiovascular diseases [6]. Resveratrol is now being used increasingly in cosmetology and dermatology owing to its anti-aging properties [7]. Although resveratrol also exhibits hypopigmenting effects, so far this property has been less noted. Resveratrol differs from other hypopigmenting agents in that it inhibits melanin synthesis by a wide variety of mechanisms. It can function not only as a direct tyrosinase inhibitor, due to its ability to serve as an alternative substrate for tyrosinase [8], but also as an indirect inhibitor, due to its ability to inhibit transcription of tyrosinase or regulate it post-transcriptionally [9,10]. In addition, it can affect keratinocytes by regulating their inflammatory processes, protecting them from oxidative damage, and repairing the basement membrane [11,12]; this, in turn, aids in regulating the function of melanocytes. Furthermore, the antioxidant activity of resveratrol may inhibit melanin formation and exert systemic effects for effective treatment of melasma, one of the representative pigmentary skin disorders. Here, we review the published literature on the topic and describe the potential of resveratrol to act as a hypopigmenting agent based on its diverse activities.

## 2. Effects on Melanocytes

Melanins, including eumelanin and pheomelanin, are pigment molecules that are produced mainly in a specialized group of cells known as melanocytes. Melanogenesis is a complex process that is regulated by numerous signaling systems (Figure 1) [13]. Various hypopigmenting agents have been identified based on their different mechanisms of action for melanogenesis in melanocytes (Table 1) [5,14].

### 2.1. Resveratrol as a Direct Tyrosinase Inhibitor

Tyrosinase is an important enzyme in melanogenesis [15], and most hypopigmenting agents are direct inhibitors of tyrosinase. To date, the drug of choice for treating pigmentary conditions has been a triple-combination cream containing 4% HQ as the main ingredient, 0.05% tretinoin, and 0.01% fluocinolone acetonide [16]. Hydroquinone, a tyrosinase inhibitor, causes the formation of oxidation products that can then induce oxidative damage of membrane lipids and proteins, and deplete glutathione [17]. It is usually used at a concentration of 4–5% in topical agents, but it is not commonly used in cosmetics because of the high frequency of adverse effects, including skin irritation [1,2].

Resveratrol is a hydroxystilbene compound that is a type of natural phenol found in grapes, red wine, berries, and other plants. It has anticancer, anti-hyperlipidemia, and anti-aging properties [18,19]. Generally, phenolic compounds are known to act as hypopigmenting agents because of their ability to serve as alternative substrates for tyrosinase [8]. L-tyrosine and other phenolic compounds, including resveratrol and HQ, commonly have a phenolic hydroxyl (-OH group). This structural similarity between phenolic compounds and the melanogenic precursor enables phenolic compounds to interact with tyrosinase and inhibit it [20,21]. Resveratrol is reported to have tyrosinase inhibitor activity in mushroom tyrosinase (Figure 1, Table 2) [22]. Compared to other hypopigmenting agents, including hydroquinone, ascorbic acid, or arbutin, resveratrol is a much more potent tyrosinase inhibitor [23]. While the half maximal inhibitory concentration (IC50) of HQ against human tyrosinase is in the millimolar range [24], that of resveratrol is 1.8 μM (Table 2) [25]. Oxyresveratrol, a naturally occurring analog, is even more potent than resveratrol with an IC50 value of 0.09 μM [25]. A concentration of only 1 μM of oxyresveratrol inhibits 50% of dopa oxidase activity (Table 2) [26].

### 2.2. Resveratrol as an Indirect Tyrosinase Inhibitor

#### 2.2.1. Resveratrol as an Inhibitor of Tyrosinase Transcription

Besides direct inhibition, tyrosinase activity in cells can be decreased through other means, such as by the reduction of tyrosinase gene transcription. In melanocytes, the transcription of genes encoding tyrosinase and tyrosinase-related protein-1 (TRP-1) is controlled by microphthalmia transcription factor (MITF) (Figure 1) [27]. MITF is a crucial transcription factor for both melanocyte proliferation and melanogenesis [28]. As MITF is regulated by the Wnt signaling pathway, entities, including cyclic adenosine monophosphate (cAMP), mitogen-activated protein (MAP) kinase pathway, and any other agents that regulate the Wnt signaling pathway, will affect MITF and, thus, melanogenesis [13,29]. For example, transforming growth factor-β1 regulates melanogenesis via extracellular signal-regulated kinase (ERK)-induced downregulation of MITF [30]. In addition, lysophosphatidic acid and C2 ceramides inhibit melanogenesis by inducing MITF degradation or by decreasing MITF expression [31,32,33].

Resveratrol is known to inhibit α-melanocyte-stimulating hormone (MSH) signaling in melanoma cells and to reduce the tyrosinases TRP-1, 2, and MITF (Table 2) [9,34,35]. Kim et al. demonstrated that autophagy induced by resveratrol suppressed α-MSH-induced melanogenesis [36]. It also prevented inflammation and oxidative stress by downregulating protein kinase C (PKC)-α (Table 2) [37]. Since oxidative processes and the PKC pathway are important in melanogenesis [38,39], resveratrol may inhibit melanogenesis by inhibiting these processes. Hagiwara et al. demonstrated that resveratrol suppressed antimycin A-mediated reactive oxygen species (ROS) production in melanocytic cells [23]. Furthermore, resveratrol can also decrease melanogenesis by regulating the MAP kinase pathway, which is another important signaling pathway in melanogenesis. As shown in Table 1, sphingosine-1-phosphate (S1P) interfered with melanogenesis via ERK-activated transcription regulation [40]. Interestingly, resveratrol stimulated S1P signaling in keratinocytes [41]. Recently we discovered that resveratrol inhibits melanogenesis through the activation of forkhead box O3 (FOXO3) without surtuin 1 (SIRT1) activation (Table 2) [42]. In our study, it was clearly shown that resveratrol inhibited melanogenesis through ERK activation followed by MITF downregulation (Figure 2). In summary, resveratrol decreases transcription of tyrosinase by regulating cAMP, PKC, and MAP kinase pathways (Figure 1).

#### 2.2.2. Resveratrol as a Post-Transcriptional Regulator of Tyrosinase

To decrease melanin formation, post-transcriptional regulation can be another way to decrease tyrosinase activity in cells. Some agents inhibit melanogenesis by increasing the degradation of tyrosinase proteins. Unsaturated linoleic acid decreases tyrosinase activity, whereas saturated palmitic or stearic acids increase tyrosinase activity [43]. Linoleic acid decreases tyrosinase levels by increasing tyrosinase ubiquitination and degradation via the proteasome [44,45]. Additionally, other agents, such as phospholipase D2, decrease melanogenesis through the same ubiquitin-mediated degradation of tyrosinase [46].

Resveratrol exerts post-transcriptional effects on tyrosinase. Retention of immature tyrosinase in the endoplasmic reticulum by resveratrol reduces the levels of fully processed tyrosinase (Table 2) [10].

### 2.3. Summary

Overall, these findings demonstrate that resveratrol is not only a direct inhibitor of tyrosinase but is also involved in transcription and post-transcriptional processing of tyrosinase. Resveratrol has diverse effects on tyrosinase, indicating that resveratrol can be a potent antimelanogenic agent.

## 3. Effects on Keratinocytes

Keratinocytes and melanocytes constitute the epidermal melanin units and closely interact with each other [47]. This close relationship between these cell types is very important in the regulation of melanogenesis [48].

### 3.1. Resveratrol as an Anti-Inflammatory Regulator

Cutaneous inflammation due to various stimuli, including UV exposure and inflammatory skin diseases, may result in hyperpigmentation [49,50]. The release of inflammatory cytokines, chemokines, arachidonic acid metabolites (leukotrienes, prostaglandins, thromboxane), as well as ROS, produced by keratinocytes, overstimulate the melanocytes and, thus, melanogenesis [51]. Therefore, anti-inflammatory agents, such as topical corticosteroids, can be used to treat pigmentary disorders [52]. Although topical corticosteroids show strong anti-inflammatory properties by inhibiting nuclear factor-κB activation [53], their long-term use is not recommended due to side effects.

Resveratrol alleviates lipopolysaccharide-induced damage in HaCaT cells, a human keratinocyte cell line, by activating PTEN/PI3K/AKT pathways. It also exhibits anti-inflammatory activity through the regulation of inflammatory mediators (Table 2) [11]. Furthermore, it is well-known that resveratrol protects against inflammatory disorders by affecting arachidonic acid metabolism [54]. By reducing inflammatory mediators produced by keratinocytes, resveratrol prevents inflammation-induced melanogenesis.

### 3.2. Resveratrol as a UV Protectant

Ultraviolet radiation simulates melanogenesis [50], as well as induces free radical generation in keratinocytes which, in turn, induces melanogenesis in melanocytes. Resveratrol has beneficial effects on redox balance not only through its direct antioxidant properties but also by the upregulation of endogenous antioxidant pathways through activation of the nuclear factor erythroid 2-related factor (Nrf2) pathway [55]. Through its direct and indirect antioxidant effects, resveratrol may prevent free radical damage of keratinocytes that, in turn, affects melanogenesis. This is supported by the protective effects of resveratrol against UV-induced and cigarette smoke-induced keratinocyte damage (Table 2) [56,57]. Thus, resveratrol prevents keratinocyte-mediated indirect stimulation of melanocytes by reducing UV damage of keratinocytes. In several in vivo studies that used the guinea pig model, resveratrol was demonstrated to inhibit UV-induced pigmentation (Table 2) [12,35].

### 3.3. Resveratrol as a Niche Modulator

Melasma is a commonly acquired hypermelanosis that affects sun-exposed areas of the skin. Although it is characterized by epidermal pigmentation, signs of photoaging, including disruption of the basement membrane and solar elastosis, are also prominent in melasma [58,59]. Basement membrane disruption may induce pigment incontinence, which manifests as a dirty skin color [60,61]. Thus, basement membrane disruption can be a target to treat melasma effectively. We tested the effects of resveratrol on skin equivalent cultures. As the skin equivalent is a three-dimensional culture system, it reflects the interactions between the epidermis and dermal stromal cells observed in skin tissues in vivo. The results showed that resveratrol increased the expression of integrin α6 at the dermoepidermal junction, as well as the number of p63-positive cells in the epidermis (Figure 3). Integrin α6 is a basement membrane protein that is related to the stemness of the epidermis [62], and p63 is a putative epidermal stem cell marker [59]. Increased integrin α6 and p63 expression indicates that resveratrol repairs the basement membrane and, thus, has rescuing effects on the stemness of interfollicular epidermal cells [63].

### 3.4. Summary

Resveratrol regulates melanocytes by affecting keratinocytes. Inflammation and oxidative damage of keratinocytes are strong melanogenic stimuli. Resveratrol regulates the inflammatory processes in keratinocytes and protects them from oxidative damage through which it prevents keratinocyte-induced melanocyte stimulation. Disruption of the basement membrane is thought to cause irregular dirty pigmentation observed in melasma. Our preliminary study showed that resveratrol restored the integrity of the basement membrane. Overall, these findings suggest that resveratrol indirectly regulates pigmentation by affecting keratinocytes.

## 4. The Antimelanoma Activity of Resveratrol

### 4.1. The Role of Melanin in Melanoma Biology

Cutaneous melanoma is a malignancy of melanocytes located in the skin. It is the most rapidly increasing malignancy in Caucasian populations that causes a high mortality rate in its advanced stages [64,65]. Recent studies have revealed that melanogenesis influences the behavior of normal and malignant melanocytes, as well as their surrounding microenvironment, through the action of melanogenesis intermediates, stimulation of aerobic glycolysis, or interaction with other metabolic pathways [66,67,68,69]. Melanin pigment has been thought to play an ambivalent role in melanoma biology. Melanin can protect normal melanocytes against noxious insults, such as radiation or cellular toxins, and can help prevent malignant transformation [66,70]. However, it may also attenuate the effectiveness of radiation from chemotherapy for malignant melanoma [70,71]. Furthermore, due to immunosuppressive, genotoxic, and mutagenic properties, melanogenesis can enhance tumor growth or induce tumor progression [66,70,71,72]. Recently, it was demonstrated that high pigmentation level is inversely correlated with the overall survival time, as well as disease-free survival time, in patients with stages III and IV melanomas [71,72]. For these reasons, melanin has been considered to be a risk factor for melanoma development and progression, where regulation of melanogenesis can be employed as one of the antimelanoma approaches.

### 4.2. Resveratrol as a Potential Antimelanoma Agent

It has been shown that resveratrol has anticancer properties, both in terms of prevention of and therapy for various cancers [73]. Its ability to prevent carcinogenesis is based on the inhibition of oxidative stress, inflammation, and cancer cell proliferation, along with the activation of cell death mechanisms. Especially in in vitro and in vivo studies, resveratrol has been found to have an antimelanoma property. Resveratrol oligomers showed a strong antimelanoma effect against SK-MEL-28 melanoma cells [74]. It also showed cell-cycle disruption and apoptosis in chemoresistant B16 melanoma [75]. Moreover, resveratrol was proven to be effective in inhibiting α-MSH-induced cancer stem cell-associated molecules, such as Wnt-1/β-catenin, c-Kit, MITF, and matrix metalloproteinase 9 (MMP-9), decreasing the cell viability, as well as suppressing the invasion of melanoma B16 cells [34]. In addition, topical application of resveratrol also inhibited UVB-induced photocarcinogenesis, including melanoma in mice models [76,77,78,79]. In conclusion, topically applicable resveratrol has the potential to not only act as a potent hypopigmenting agent, but also as a chemopreventive or therapeutic agent against melanoma.

## 5. Potential to be Used as a Promising Topical Hypopigmenting Agent

Resveratrol is being increasingly used in cosmetology and dermatology [7]. Resveratrol and related chemicals have shown significant hypopigmenting effects in animal models [9,12,35]. In a clinical study, a hybrid topical compound of resveratrol and glycolic acid significantly improved human skin pigmentation [80,81]. Topical resveratryl triacetate, a prodrug of resveratrol, also exhibited a significant reduction of UV-induced pigmentation as compared to the control (Table 2) [82].

The combined use of the two agents with different mechanisms of action can also have additive effects. 4-*n*-Butylresorcinol is a strong tyrosinase inhibitor that is not involved in transcriptional or post-translational regulation of tyrosinase. When we tested the combined effects of 4-*n*-butylresorcinol and resveratrol, we observed synergistic effects [83]. Since resveratrol has very diverse effects on melanogenesis, its combination with any other hypopigmenting agents may have additive effects.

### 5.1. Bioavailability of Resveratrol Following Topical Application

Resveratrol can penetrate and accumulate in the skin. Topically applied resveratrol penetrates the skin in a gradient fashion and is able to maintain its antioxidant and anti-inflammatory efficiency in the skin [84,85]. Resveratrol in aqueous buffers of lower pH values exhibits better permeability and deposition than in buffers of higher pH or oil-based chemicals [86]. To increase the efficacy of the topical delivery of resveratrol, numerous forms of its formula have been investigated, such as in the form of solutions, gels, emulsions, and liposomes [86,87,88,89,90]. Once delivered through the skin, resveratrol undergoes glucuronidation [85]. While orally administered resveratrol rapidly decreases in the plasma because of extensive metabolism, topically applied resveratrol can be detected in the plasma for a long time, suggesting much better bioavailability as compared to oral administration [90].

### 5.2. Toxicity of Resveratrol Following Topical Application

Although only one case of contact dermatitis has been reported [91], topical resveratrol is reported to not induce skin irritation in most studies [82,87,90,92]. Topically applied resveratrol alleviated both chemical-induced irritant contact dermatitis [92] and DNFB (2,4-dinitro-1-fluorobenzene)-induced contact hypersensitivity in animal studies [93]. The anti-inflammatory effect of resveratrol may contribute to its low irritative potential. Topical resveratrol also protects the skin from UVB-induced skin inflammation [76] and UVB-mediated skin carcinogenesis [78], which are desirable characteristics in topical agents for the skin.

Resveratrol is not only a direct inhibitor of tyrosinase but also a substrate for it [8,94]. Hypopigmenting agents that act as substrate for tyrosinase have the potential to lead to leukoderma through their tyrosinase-catalyzed oxidation to cytotoxic ortho-quinones [95,96,97]. Rhododendrol is a representative example of this process; it has been demonstrated that rhododendrol-induced melanocyte cytotoxicity is enhanced by UVB exposure through the generation of oxidative stress [98,99]. Since resveratrol acts as both a substrate and an inhibitor of tyrosinase, it may induce leukoderma, theoretically. However, as mentioned above, resveratrol was reported to inhibit UVB-induced skin inflammation and oxidative stress. Moreover, in contrast to rhododendrol, suppression of pigmentation by resveratrol is not related to melanocyte toxicity [100,101]. Therefore, although further investigations are necessary to confirm the leukoderma-inducing potential of resveratrol, it is thought to have a low chance of inducing leukoderma.

## 6. Conclusions

Recently, melanocyte biology has made remarkable progress. However, the pathogenic mechanisms underlying acquired hyperpigmentation are not completely understood. As summarized in this review, resveratrol is a direct, as well as an indirect, tyrosinase inhibitor. It can also affect keratinocytes by regulating inflammatory processes and UV-induced oxidative damage, and is also involved in restoring the basement membrane and stemness of keratinocytes. Resveratrol is being increasingly used in the fields of cosmetology and dermatology. It is a promising agent for treatment of pigmentary disorders, considering its diverse effects on melanogenesis. However, despite abundant laboratory evidence, there have been scarce data from human studies. Further clinical trials to prove its efficacy and safety are necessary.

## Figures and Tables

**Figure 1 ijms-20-00956-f001:**
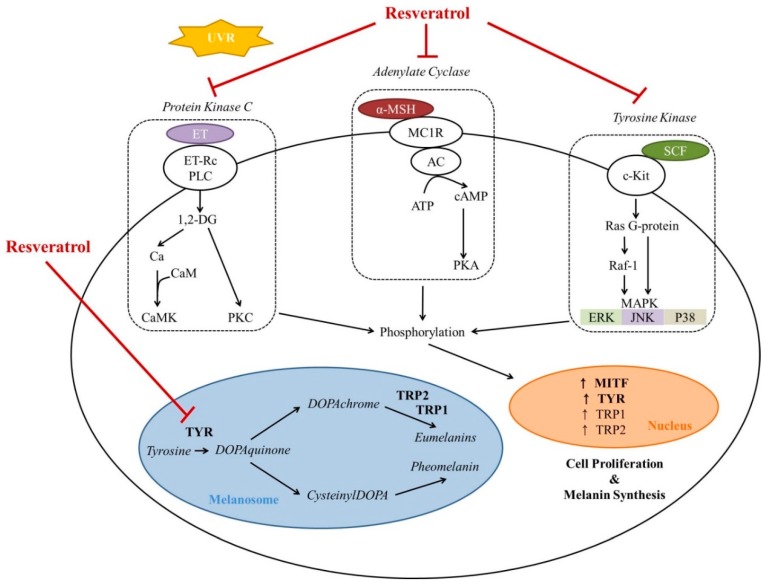
Schematic illustration of the effect of resveratrol on melanogenesis and related signaling pathways in melanocytes. The cyclic AMP-dependent protein kinase (PKA), protein kinase C (PKC), and mitogen-activated protein kinase (MAPK) signaling pathways induce the activation of transcription factors for several genes related to melanogenesis and lead to melanocyte proliferation and melanin synthesis. Resveratrol not only inhibits these signaling pathways, but also directly inhibits tyrosinase activity. AC = adenylate cyclase; CaMK = Ca^2+^/calmodulin-dependent protein kinase; DG = diacylglycerol; ERK = extracellular signal-regulated kinase; ET = endothelin; JNK = c-Jun N-terminal kinases; MITF = microphthalmia transcription factor; MSH = melanocyte-stimulating hormone; PLC = phospholipase C; Rc = receptor; SCF = stem cell factor; TRP = tyrosinase-related protein; TYR = tyrosinase; and UVR = ultraviolet radiation.

**Figure 2 ijms-20-00956-f002:**
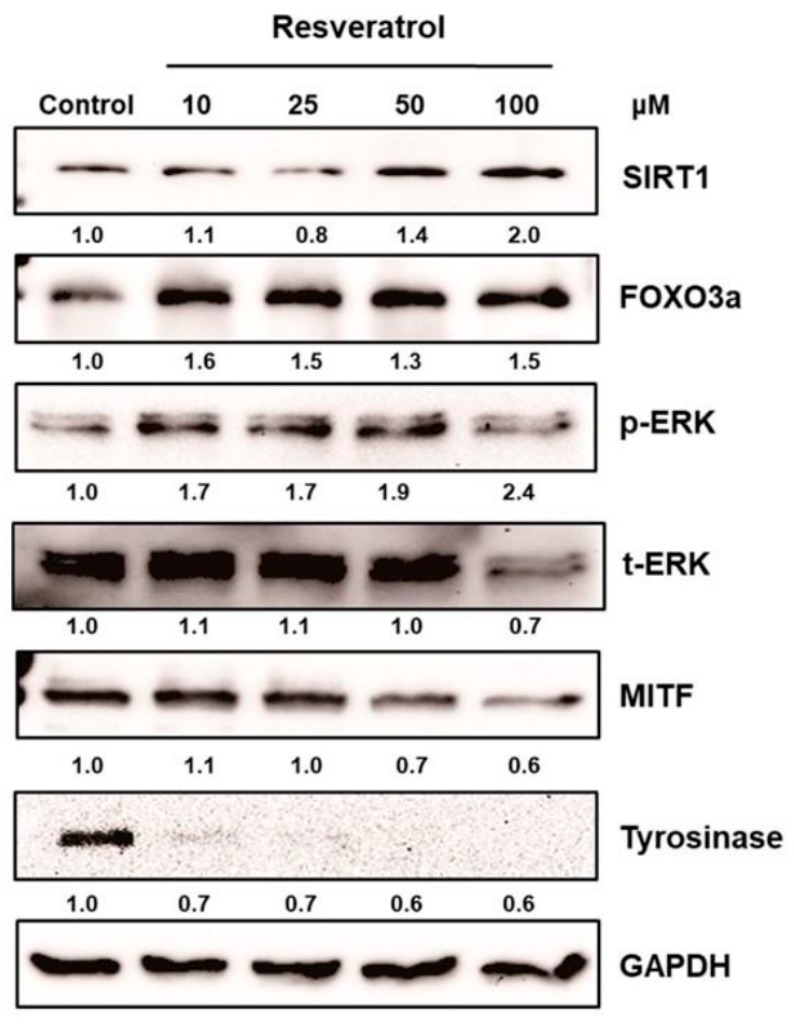
The activation of MAPK and the downregulation of MITF by resveratrol. Normal human melanocytes were treated with 10–100 μM of resveratrol for 24 h. Following this, the levels of ERK, MITF, tyrosinase, sirtuin 1 (SIRT1), and FOXO3a were investigated. Resveratrol treatment for 24 h effectively increased phosphorylation of ERK and decreased the levels of MITF and tyrosinase. The levels of SIRT1 and FOXO3a also increased in normal human melanocytes after treatment with resveratrol (adopted from [42] with permission).

**Figure 3 ijms-20-00956-f003:**
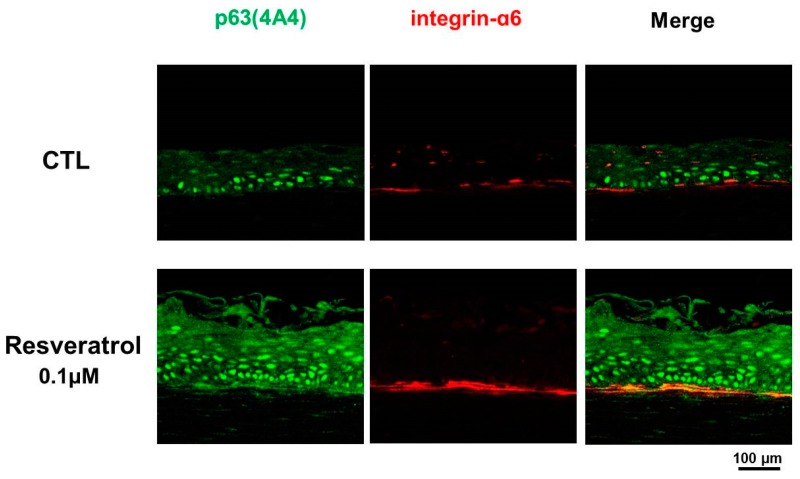
Immunohistochemical study on skin equivalents treated with resveratrol. Resveratrol-treated skin equivalents showed increased staining of integrin α6 and an increased number of p63-positive cells (green: p63 staining, red: integrin α6 staining, magnification = 200×, scale bar is 100 μm).

**Table 1 ijms-20-00956-t001:** Classification of hypopigmenting agents based on reported mechanisms of action.

Mechanisms of Action	Hypopigmenting Agent
**Before melanin synthesis**	
Regulation of tyrosinase transcription	TGF-*β*1, TNFα, IL-1α, IL-1*β*, IL-6, Lysophosphatidic acid, C2-Ceramides, Sphingosine-1-phosphate, Sphingosylphosphorylcholine, Tretinoin
Inhibition of tyrosinase maturation	Glucosamine, Tunicamycin, Glycosphingolipid, Calcium D-pantetheine-S-sulfonate
**During melanin synthesis**	
Inhibition of tyrosinase activity	Hydroquinone, Arbutin, Kojic acid, 4-*n*-Butylresorcinol, Phenolic compounds, 4-Hydroxyanisole, Methyl-gentisate, 4-S-CAP & derivatives, Ellagic acid, Oxyresveratrol, Resveratrol, Aloesin, Azelaic acid, Zinc
**After melanin synthesis**	
Post-transcriptional control of tyrosinase	Linoleic acid, α-Linolenic acid, Phospholipase D2
Inhibition of melanosome transfer	Niacinamide (Vitamin B3), serine protease inhibitors, lecthins and neoglycoproteins, RW-50353, soybean.milk extracts
Regulation of melanocyte environment	Corticosteroids, Glabridin
Antioxidants	*α*-Tocopherol, Ascorbic acid, 6-Hydroxy-3,4-dihydrocoumarins, *α*-Lipoic acid, Methimazole, Phenol/catechol, Ascorbic acid palmitate, Decursin, L-α TF, VC-PMG, Thioctic acid

TGF = transforming growth factor, TNF = tumor necrosis factor, IL = interleukin, 4-S-CAP = 4-S-cysteaminylphenol, TF = tocopherol ferulate, VC-PMG = magnesium L-ascorbyl-2-phosphate.

**Table 2 ijms-20-00956-t002:** Summary of biological activities of resveratrol.

Study Models	Biological Activities of Resveratrol	References
**Enzyme inhibition**		
	inhibits mushroom tyrosinase	[22,26]
	inhibits human tyrosinase	[25,10]
**Cell culture**		
	reduces melanin production in B16 murine melanoma cells	[22,25,34]
	reduces melanin production in human epidermal melanocytes	[25]
	reduces MITF and tyrosinase transcription in B16 murine melanoma cells	[34,35]
	reduces MITF and tyrosinase in human epidermal melanocytes	[9,42]
	induces autophagy and reduces α-MSH-induced melanogenesis in melan-A cells	[24]
	downregulates PKC-α in lung epithelial A549 cells	[37]
	increases phosphorylation of ERK in human epidermal melanocytes	[42]
	reduces the post-transcriptional process of tyrosinase in human epidermal melanocytes	[10]
	reduces inflammatory injury in HaCaT cells	[11]
	reduces UVB-induced injury in HaCaT cells	[56]
	prevents oxidative stress-induced injury in human keratinocytes	[57]
**Reconstructed skin model**		
	reduces melanin production in a reconstituted human skin model	[25]
**Animal model**		
	reduces skin pigmentation in Yucatan swine, 1% topical resveratrol for 8 weeks	[9]
	reduces UVB-induced skin pigmentation in guinea pig, 1% topical resveratrol for 2 weeks	[35]
	reduces UVB-induced skin pigmentation in guinea pig, topical application of callus of resveratrol-enriched rice for 15 days	[12]
**Human clinical trials**		
	enhances depigmentation after UV-induced tanning (*p* < 0.05), topical 0.4% resveratrol triglycolate vs. control cream, 22 subjects for 8 weeks	[81]
	decreases hyperpigmented spots on the face (*p* < 0.05), topical 0.4% resveratrol triacetate vs. control cream, 21 subjects for 8 weeks	[82]

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
