# Peer review of "Resveratrol as a Multifunctional Topical Hypopigmenting Agent"

_ijms, 2019, doi:10.3390/ijms20040956_

Reviewer 1 Report

The authors have addressed satisfactorily the issues raised by the Reviewers. As a result, the review is more informative and useful for the readership. However, a few minor issues remain and must be addressed before the manuscript can be accepted for publication:

1. In line 30, "contact more dermatitis" should be replaced by "contact dermatitis"

2. In line 31, the sentence should start by "It can also induce..."

3. In Fig. 3, uM should be written correctly (μM)

4. In section 4.2, "proven" (appears twice) should be replaced by "shown" or "found"; The last sentence should start by "In conclusion, ...".

Author Response

Thank you for your careful corrections. The manuscript was revised according to your suggestions now.

1. It was replaced by “contact dermatitis”.

2. It was corrected as you suggested.

3.  It was corrected.

4. They were all corrected according to your suggestions.

Reviewer 2 Report

The Authors have made much effort to improve the already well-written review..

1. The names of journals in the reference list are still not uniformly presented - abbreviations besides full titles and fully capitalized titles besides only the first word of the title capitalized. Please correct them now or on proof.

2. Table 2 - the original references are not given. Would it add much text if the original references concerning the data in Tab 1 were cited in the reference list? I realize it is not the direct topic of the review, but the reader may intend to compare some facts and/or look for common mechanism... It may expand the "printed" (virtually) version of the paper by ca. 1 page. It is the matter to decide for authors and editors, so I do not press but leave it at their discretion.

Author Response

1. The title of ref 58 and 63 were now corrected. We will check them on proof again.

2. Thank you for raising this query. It must be regarding Table 1 not Table 2, isnt’ it? We fully understand and agree with your point. However, the amount of original references for Table 1 is massive and citation of all these references might make our manuscript somewhat redundant. In addition, as you mentioned, it is not the main topic of the review. Thus, we would be better to leave it as it is.

This manuscript is a resubmission of an earlier submission. The following is a list of the peer review reports and author responses from that submission.

Round  1

Reviewer 1 Report

This paper by Na, Shin et al. reviews the current status of resveratrol as a multifunctional topical hypopigmenting agent. The review nicely summarizes classification of hypopimenting agents based on reported mechanisms and biological activities of resveratrol. It is easy to read and is comprehensive with regard to the activities of resveratrol. Nevertheless, this review needs to be improved in the following major and minor points.

Major point:

Although resveratrol may directly inhibit mushroom and human tyrosinases, it also acts as a substrate for various tyrosinases as a para-substituted phenol (see references 1-4, 4=Ref. 13 of the review). Therefore, it is likely that resveratrol acts as both substrate and inhibitor of tyrosinases, thus leading to hypopigmentation in vitro and in vivo. This situation is similar to that of rhododendrol, a hypopigmenting agent that caused leukoderma among many users (reference 5). Rhododendrol acts as not only inhibitor but also substrate of tyrosinases and induced leukoderma through tyrosinase-catalyzed oxidation to cytotoxic ortho-quinone (reference 6). A similar possibility should be mentioned for the action of resveratrol.  

Minor points:

1) Hagiwara et al. (reference 7) examined the effects of resveratrol on mushroom tyrosinase, expressions of tyrosinase and TRP1, and ROS production in comparison with other hypopigmenting agents. This paper should be cited.

2) Table 1: Style is not consistent. Examples are:

a) ‘Tunicamycin’ should be ‘tunicamycin’.

b) ‘4-Hydroxy-anisole’ should be ‘4-Hydroxyanisole’.

c) The list of compounds in ‘inhibition of tyrosinase activity’ is identical to that in ‘inhibition of melanosome transfer’. Is this correct?

d) In ‘Antioxidants’, there are ‘6-hydroxy-3,4-dihydrocoumarins’ and ‘Hydroxycoumarins’. Is this OK?

e) A few reviews should be cited in regard to the list of Table 1.

3) There are a few typographical errors including:

a) Line 50: ‘regarding to melanogenesis’ should be ‘regarding melanogenesis’.

b) Line 54: Ca2+ should be Ca2+.

c) Line 78: ‘with_IC50’ should be ‘with IC50’.

d) Line 136: ‘result hyperpigmentation’ should be ‘result in hyperpigmentation’.

e) Line 196, Title: ‘Bioavailability of Resveratrol from Topical Application’.

f) Line 207, Title: ‘Toxicity of Resveratrol from Topical Application’.

g) Table 2, Enzyme inhibition: ‘inhibits’ but not ‘Inhibits’.

h) References 32, 50, and 51: Only the first word of the title starts with upper case letter.

1. Gonzalvez, A.G. et al. (2012). Spectroscopy and kinetics of tyrosinase catalyzed trans-resveratrol oxidation. J. Physical Chem. B, 116, 2553-2560.

2. Lee, S.H. et al. (2016). Using tyrosinase as a monophenol monooxygenase: a combined strategy for effective inhibition of melanin formation. Biotech. Bioeng. 113, 735-743.

3. Lee, N., et al. (2015). Heterologous expression of tyrosinase (MelC2) from Streptomyces avermitilis MA4680 in E. coli and its application for ortho-hydroxylation of resveratrol to produce piceatannol. Appl. Microbiol. Biotech., 99, 7915-7924.

4. Satooka, H., and Kubo, I. (2012). Resveratrol as a kcat type inhibitor for tyrosinase: potentiated melanogenesis inhibitor. Bioorg. Med. Chem. 15, 1090-1099.

5. Nishigori, C. et al. Nishigori, C. (2015). Guide for medical professionals (i.e., dermatologists) for the management of rhododenol-induced leukoderma. J. Dermatol. 42, 113-128.

6. Ito S., and Wakamatsu, K. (2018). Biochemical mechanism of rhododendrol-induced leukoderma. Int. J. Mol. Sci. 19, E552.

7. Hagiwara K, et al. (2016). Biochemical effects of the flavanol-rich lychee fruit extract on the melanin biosynthesis and reactive oxygen species. J.

Dermatol. 43, 1174-1183.

Author Respons

Thank you for your valuable comments and kind corrections. The manuscript was revised according to your suggestions now.

Major point:

As you mentioned, hypopigmenting agents including rhododendrol that act as a substrate for tyrosinase and form ortho-quinone have potential to lead leukoderma (Nishigori, C. et al. J. Dermatol. 42, 113-128, Ito S et al. J Dermatol Sci. 2015;(80);18-24, Ito S. et al. Int J Mol Sci. 2018; 19, E552). And it was demonstrated that rhododendrol-induced melanocyte cytotoxicity is enhanced by UVB exposure though generation of oxidative stress (Goto N, et al. Exp Dermatol. 2018;27:754-762). Since resveratrol acts as both substrate and inhibitor of tyrosinase, it may induce leukoderma theoretically. However, resveratrol was reported to inhibit UVB-induced skin inflammation and oxidative stress in a mice study (Fujimura AT. et al. J Nat Prod. 2016;79:1329-38). Thus, it is necessary to test whether the treatment of resveratrol with tyrosinase may generate ortho-quinone to evaluate the potential to lead leukoderma by resveratrol. These contents were included in ‘4.2. Toxicity of resveratrol from topical application’.

Minor points:

1) Thank you for pointing this out. This paper was now cited in the article.

2) Table 1 was revised according to your suggestion. Thank you.

a) Corrected

b) Corrected

c) It was our mistake. Now, the list of ‘inhibition of melanosome transfer’ was corrected.

d) They are identical compounds and duplicated. ‘Hydrocoumarins’ was deleted now.

e) Several reviews (Briganti S, et. al. Pigment Cell Res. 2003 Apr;16(2):101-10., and Kim H. et al. Ann Dermatol. 2012 Feb; 24(1): 1–6) regarding Table 1 were cited in the manuscript.

3) Thank you for pointing this out. They were all corrected.

Reviewer 2 Report

This is a comprehensive, well written review on resveratrol, an import and polyphenolic compound, however, of a lesser known aspects of its activity as an atimelanogenic agent. The paper would be in general ready for publication but for a couple of technical errors and some suggestions to expand the review by a couple of facts, if possible.

1.  The most important is the neglection of potential anti-melanoma activity of reseveratrol, coupled with its  antimelanogenic activity. Melanin is an important factor of melanoma induction and progression, which should find its reflex in this review if the authors want to enrich the paper with a practical aspect. As a “bleaching” agent, the substance would possess a practical value, but potential antimelanoma and chemopreventing activity deserves the particular attention. Melanin is of an ambivalent importance in melanoma biology, recently even treated more as a risk factor. This is because melanin may be a potential source of disregulation of tumor metabolism and anti-tumor immunity, the potential factor protecting the tumor from some modes of treatment, and because regulation of melanogenesis in part overlaps with the pathways which, if mutated, may lead to melanoma induction or progression, including the described MAPK, PI3K/Akt/mTOR, c-KIT, Wnt/b-catenin etc. pathways. The potential anti-melanoma aspect of the anti-melanogenic activity of resveratrol should be separately accented in this review. There is a substantial literature on this topic. The authors may limit themselves to those papers, which especially concern the resveratrol anti-melanoma effects concerning melanogenesis regulation.

2.  The paper may be enriched with some general information on the role of melanin, in particular in melanoma genesis. Please consider  your referring to such papers as:  Slominski et al. 2004, PMID: 15383650; 2014; PMID: 24997364; 2015, PMID: 25496715; Brozyna et al., 2013, PMID: 23791398; Cascinelli et al., 2000, PMID: 11099112; and: Ortonne, J.- P. and Ballotti, R. (2000) Melanocyte biology and melanogenesis: what's new? J Dermatol Treatment 11, S15-S26, and others.

Minor points

3.   You may also consider NO and NOS activity (primarily NOS1, either in melanocytes or in keratinocytes), as a factor affectable by resveratrol, and itself affecting melanogenesis via cGMP/PKG pathway (See e.g. Slominski et al., 2004, op. cit., and the respective papers cited there).

4.   Table 1 – please include also the pigmentation effects of oral or topical zinc, a well-known dermatologic factor which reveals versatile interference with melanogenesis.

5.  Line 48 – not only melanocytes have been proven to produce melanin, so please change for “mainly in specialized group of cells…”

6.   Figure 1 – as the PKA pathway seems the most important in melanogenesis regulation, please either indicate it with a thicker line, or place this pathway in the central part of the Figure.

7.  L. 62, 80, 115, 148, do not capitalize “a” in the chapter title (l. 115 – insert “a”}

8.  Please supplement the review with the chemical formula for resveratrol, L-Tyr and HQ, to exhibit the structural similarity.

9.   Line 86 – please cite Vachtenheim & Borovansky 2011 PMID: 20201954

10.  Line 92 – How is resveratrol “inhibiting MSH signaling”? Is it an agonist of Mc1r?, an inhibitor of the expression of POMC  of the proteolytic edition of MSH? Of adenylate cyclase? Btw – here it there is something said about melanoma (see p. 1)!

11.  Please mind that FOX stands for “Forkhead box” not “Forehead”. Please change in line 102 and 112.

12.  L 145 – “metabolism” (singular)

13.  Tab 2 – besides [24] – “reduces” (instead of “reduce”)

14.   Please re-format the references – in particular unify the journal titles, fully capitalized (according to the valid IJMS format) 

Author Response

Thank you for your valuable comments and kind corrections. The manuscript was revised according to your suggestions now.

1. Thank you for raising this query. As you mentioned, melanin pigment has been thought to play an ambivalent role in melanoma biology. Melanin can protect normal melanocytes against noxious insults such as radiation or cellular toxins and prevent malignant transformation (Slominski & 1 of Slominski). However, inversely, it also may attenuate effectiveness of radiation of chemotherapy for malignant melanoma (Slominski & 6 of Slominski). Also due to immunosuppressive, genotoxic and mutagenic properties, melanogenesis can enhance tumor growth or induce tumor progression (slominski, 1, 6, 10 of
slominski). Recent studies demonstrated that high pigmented level is inversely correlated with overall survival time and disease-free survival time in patients with stages III and IV melanoms (6,7,10 of slominski). In addition, as you commented, several signaling pathways involved in melanogenesis correspond to those involved in melanoma induction or progression. For these reasons, regulation of melanogenesis can be one of the anti-melanoma approaches.
It has been proven that resveratrol has an anti-cancer property both for the prevention and therapy of various cancers (IJMS 2017). The ability to prevent carcinogenesis includes the inhibition of oxidative stress, inflammation, and cancer cell proliferation, and the activation of cell-death mechanisms. Especially, in in vitro and in vivo studies, resveratrol has been proven to have an antimelanoma property. Resveratrol oligomer showed a strong antimelanoma effect against SK-MEL-28 melanoma cells (Moriyama, H.;Biol. Pharm. Bull. 2016, 39, 1675). It also showed cell-cycle disruption and apoptosis in chemoresistant B16 melanoma (J. Cell. Biochem. 2010, 110, 893–902). In addition, topical application of resveratrol also inhibited UVB-induced photocarcinogenesis in cluding melanoma in mice models (202-205 of Ko). Conclusively, topical resveratrol has a potential as not only a potent hypopigmenting agent but also as chemopreventive agent for melanoma. These contents including the relationship between melanin and melanoma genesis and anti-melanoma effect of resveratrol were included in the article.

2. Please refer to the answer to #1.

Minor points

3. Thank you for pointing this out. As you mentioned, nitric oxide (NO) stimulates melanogenesis via cGMP/PKG pathway. However, resveratrol generally stimulates NO production in various tissues (Molecules. 2014 Oct 9;19(10):16102-21, BMC Complement Altern Med. 2017 May 8;17(1):254, PLoS One. 2014 Oct 14;9(10):e110487). Even though resveratrol showed protective effects against the toxicity induced by nitric oxide in human skin, its protective action is not likely to be associated to the inhibition of NO synthase (PLoS One. 2010; 5(9): e12935). So we did not include NO related contents in the manuscript.

4. According to your suggestion (Exp Dermatol. 2004 Aug;13(8):465-71., Br J Dermatol. 2006 Jul;155(1):39-49) zinc was included in Table 1.

5. We changed the sentence according to your suggestion.

6. Now, PKA pathway in Figure 1 was placed in the center of the figure.

7. They were corrected.

8. L-tyrosine, HQ, and resveratrol commonly have a phenolic O-H group. This structural similarity between HQ/resveratrol and melanogenic precursor enables HQ/resveratrol’s interaction with tyrosinase and inhibition of tyrosinase (Int. J. Mol. Sci. 2009, 10, 4066-4087, J Enz Inhibit Med Chem. 2017, 32, 403-425). These contents were now included in the article.

9. That article was cited now.

10. Further investigation is necessary to elucidate the exact mechanism of α-MSH signaling inhibition of resveratrol. Chen, et. al. (Evid Based Complement Alternat Med 2013, 2013, 632121.)demonstrated that stimulation of melanoma B16 cells with α-MSH leads increase of melanin production to be accompanied with the upregulation of cancer stem cell associated molecules (Wnt-1/β-catenin, c-Kit, MITF, and MMP-9) and invasion ability. And resveratrol was effective in inhibiting the abovementioned molecule expression, decreasing the cell viability, as well as suppressing the invasion of melanoma B16 cells. This implicates resveratrol may have potential to be developed as a novel therapeutic agent against CSCs of melanoma. These contents were added in the article.

11. They were corrected.

12.  It was corrected.

13.  It was corrected.

14. Re-format of reference style was done.

Reviewer 3 Report

The manuscript by Na et al. provides a comprehensive review of the effects of resveratrol not only in melanocytes but also in keratinocytes. Moreover, it lists the known hypopigmentation agents. Despite being very focused on reseveratrol and therefore of limited scope, it is a useful resource. However, the text needs to be reviewed to address the following shortcomings:

1. In the Introduction, the authors should be more specific about the adverse effects of HQ (line 30) and the meaning of the direct vs. indirect effects of resveratrol (line 39). It is true that these are explained later, but the reader should not be left wondering. Moreover, the sentence in line 39 must be referenced.

2. Also in the Introduction, the authors should be more specific about what they mean by "...it can affect keratinocytes..." (line 40). How does it affect keratinocytes?

3.  The last sentence of the Introduction is awkward and does not say anything concrete. Therefore, it should be removed.

4 . In line 47 the authors should say what they mean by "melanins". Are they referring to eumelanin and pheomelanin? If yes, this has to be explicit.

5 . The legend of Fig. 1 is very poor and the schematic needs to be explained. Moreover, the authors did not define the acronym "SCF".

6 . In line 129, the term "melanogenetic" is awkward. Do the authors mean "melanogenic"?

7 . In section 3.2, what is the animal model the authors refer to (line 157)? This must be specified.

8 . The legend of Fig. 3 is also very poor and does not contain any technical details as it should. The legend of Fig. 2 can also be improved, albeit not being as poor.

9.  In line 182, the term "clearly showed" must be downplayed significantly, since the authors do not show or mention any barrier functional assay that would more clearly show if the basement barrier is disrupted or not.

10.  ln lines 177 and 183, the authors refer to the modulation of keratinocytes. What do they mean? This has to be made clear.

11 . The authors should add a subtopic in section 4 (it would be 4.3), where they discuss the possible side effects of resveratrol.

Finally, several typos and minor mistakes that are too many to be listed, should be corrected.

Author response

Thank you for your valuable comments and kind corrections. The manuscript was revised according to your suggestions now.

1.Concrete examples of adverse effect of HQ were added in the article. In addition, line 39 was explained more specifically and referenced.

2. The effect of resveratrol on keratinocytes were briefly added in introduction now.

3. It was removed now according to your suggestion.

4. Yes, “melanins” means eumelanin and pheomelanin. It was explicit now.

5. Thank you for your comments. The legend of Fig.1 was explained more in detail and the definition of “SCF” was added.

6. “Melanogenetic” was corrected to “melanogenic”.

7. The studies commented in line 157 used guinea pig model. This was specified now in the article.

8. The legends of Fig. 2 and 3 were supplemented.

9. As you mentioned, we showed the structural integrity of basement membrane immunohistochemically and did not perform functional assay. So the term “clearly” was removed.

10. Resveratrol regulates the inflammatory process of keratinocytes and protects keratinocytes from oxidative damage, through which it prevents keratinocyte-induced melanocyte stimulation. In addtion, resveratrol repairs the basement membrane and has rescuing effects on the stemness of interfollicular epidermal cells. As you mentioned, though, the term ‘modulation’ is some what not appropriate in this situation. So, we changed ‘modulation’ to ‘affecting’.

11. Thank you for your comment. The contents about possible side effect of leukoderma by resveratrol were added in section 4 as you suggested.

Finally, Thank you for your comments. Mistakes in this article were corrected, and professional English editing was done additionally.